# Chemical Composition and Antimicrobial Activity of New Honey Varietals

**DOI:** 10.3390/ijerph20032458

**Published:** 2023-01-30

**Authors:** Magdalena Kunat-Budzyńska, Anna Rysiak, Adrian Wiater, Marcin Grąz, Mariola Andrejko, Michał Budzyński, Maciej S. Bryś, Marcin Sudziński, Michał Tomczyk, Marek Gancarz, Robert Rusinek, Aneta A. Ptaszyńska

**Affiliations:** 1Department of Immunobiology, Institute of Biological Sciences, Faculty of Biology and Biotechnology, Maria Curie-Skłodowska University, Akademicka 19 Str., 20-033 Lublin, Poland; 2Department of Botany, Mycology, and Ecology, Institute of Biological Sciences, Faculty of Biology and Biotechnology, Maria Curie-Skłodowska University, Akademicka 19 Str., 20-033 Lublin, Poland; 3Department of Industrial and Environmental Microbiology, Institute of Biological Sciences, Faculty of Biology and Biotechnology, Maria Curie-Skłodowska University, Akademicka 19 Str., 20-033 Lublin, Poland; 4Department of Biochemistry and Biotechnology, Institute of Biological Sciences, Faculty of Biology and Biotechnology, Maria Curie-Skłodowska University, Akademicka 19 Str., 20-033 Lublin, Poland; 5Urban Artistic Apiary, Centre for the Meeting of Cultures, Plac Teatralny 1 Str., 20-029 Lublin, Poland; 6Department of Pharmacognosy, Faculty of Pharmacy with the Division of Laboratory Medicine, Medical University of Bialystok, Mickiewicza 2a Str., 15-230 Białystok, Poland; 7Faculty of Production and Power Engineering, University of Agriculture in Krakow, Balicka 116B, 30-149 Krakow, Poland; 8Institute of Agrophysics, Polish Academy of Sciences, Doświadczalna 4, 20-290 Lublin, Poland

**Keywords:** honey, antibacterial, antifungal, phenolic acids, lysozyme-like activity, hydrogen peroxide

## Abstract

Due to a widespread occurrence of multidrug-resistant pathogenic strains of bacteria, there is an urgent need to look for antimicrobial substances, and honey with its antimicrobial properties is a very promising substance. In this study, we examined for the first time antimicrobial properties of novel varietal honeys, i.e., plum, rapeseed, Lime, *Phacelia*, honeydew, sunflower, willow, and multifloral-P (*Prunus spinosa* L.), multifloral-AP (*Acer negundo* L., *Prunus spinosa* L.), multifloral-Sa (*Salix* sp.), multifloral-Br (*Brassica napus* L.). Their antimicrobial activity was tested against bacteria (such as *Escherichia coli*, *Bacillus circulans*, *Staphylococcus aureus*, *Pseudomonas aeruginosa*), yeasts (such as *Saccharomyces cerevisiae* and *Candida albicans*) and mold fungi (such as *Aspergillus niger*). In tested honeys, phenolic acids constituted one of the most important groups of compounds with antimicrobial properties. Our study found phenolic acids to occur in greatest amount in honeydew honey (808.05 µg GAE/g), with the highest antifungal activity aiming at *A. niger*. It was caffeic acid that was discovered in the greatest amount (in comparison with all phenolic acids tested). It was found in the highest amount in such honeys as phacelia—356.72 µg/g, multifloral (MSa) and multifloral (MBr)—318.9 µg/g. The highest bactericidal activity against *S. aureus* was found in multifloral honeys MSa and MBr. Additionally, the highest amount of syringic acid and cinnamic acid was identified in rapeseed honey. Multifloral honey (MAP) showed the highest bactericidal activity against *E. coli*, and multifloral honey (MSa) against *S. aureus*. Additionally, multifloral honey (MBr) was effective against *E. coli* and *S. aureus*. Compounds in honeys, such as lysozyme-like and phenolic acids, i.e., coumaric, caffeic, cinnamic and syringic acids, played key roles in the health-benefit properties of honeys tested in our study.

## 1. Introduction

Recently, due to the growing resistance of microorganisms to many antibiotics, attention has been paid to agents of natural origin with antimicrobial effects. Honey can be included among such substances [1,2]. The main ingredients of honey are simple sugars: fructose and glucose (about 70–80%). In addition, honey contains other sugars, e.g., maltose, sucrose and polysaccharides such as dextrins, the amount of which exceeds 10% in honeydew honey. Other ingredients of honey include proteins (about 3%), primary enzymes that come from honey bees’ bodies, as well as plant enzymes and proteins found in nectar and pollen. These include invertase, lactase, α- and β-amylase, glucose oxidase, catalase and phosphatase. In addition, honeys contain water (usually approximately 18%) and minerals, mainly potassium, small amounts of vitamins (C, H, PP, and group B) and organic acids such as gluconic, malic, citric, butyric, formic, lactic, succinic, pyroglutamic and others. The chemical composition of honey depends on the species of plants foraged by bees to produce it [1,3].

The antibacterial and antifungal properties of honey depend on its chemical, physical and biological factors. These factors differ significantly for different varieties of honey and determine various properties of honeys, e.g., Tualang honey has been proved to have impact on oral diseases, acacia honey has antiproliferation potential and manuka honey has antibacterial activity against *Helicobacter pylori* [1]. Physical factors include the low pH of honey and high osmotic pressure, which are the result of a high concentration of sugars, which in turn affects the elimination and inhibition of the growth of most microorganisms. The presence of phenolic compounds (phenolic acids and flavonoids) and hydrogen peroxide produced in the enzymatic reaction are classified as chemical factors. Hydrogen peroxide is formed as a by-product in the oxidation of glucose to gluconolactone catalyzed by glucose oxidase in the presence of atmospheric oxygen. It shows strong activity against Gram-positive bacteria, e.g., *Staphylococcus aureus*, Gram-negative bacteria such as *Pseudomonas aeruginosa* and mold fungi (*Aspergillus niger*) [4,5]. Examples of phenolic acids found in honey include, among others, caffeic, gallic, syringic and *p*-coumaric acids. The presence of flavonoids, i.e., apigenin, kaempferol, quercetin or luteolin, has been detected in honeys originating in Poland [6]. The biological factors that affect the development of microorganisms in honey include lysozyme-like activity and peptide defensin-1 [7,8,9]. Lysozyme is a protein with enzymatic activity whose molecular weight is about 14.4 kDa. Its main function is the lysis of cell walls of Gram-positive bacteria due to the content of *N*-acetylmuramic acid. This enzyme is less active against Gram-negative bacteria [10]. Defensin-1 peptide with a molecular weight of about 5 kDa is secreted by the hypopharyngeal glands of honey bees. It is characterized by activity against Gram-positive (*Bacillus subtilis*, *S. aureus*) and Gram-negative bacteria (*Escherichia coli*, *Burkholderia cepacia*) [7,11,12].

Many authors report that varietal honeys, e.g., manuka honey, have antimicrobial activity against antibiotic-resistant bacteria, including *P. aeruginosa*, while others, such as Agastache honey, have been shown to be more effective in inhibiting biofilm formation by methicillin-resistant *S. aureus* [13,14]. In addition to its antimicrobial activity, honey can also provide protection against oxidative stress and skin cancer, endometrial cancer and bladder cancer development [1,15,16,17].

Due to the broad spectrum of honey properties, this research was aimed at determining a type of honey, the content of its bioactive substances and its antimicrobial properties against bacteria (such as *Escherichia coli*, *Bacillus circulans*, *Staphylococcus aureus*, *Pseudomonas aeruginosa*), yeasts (such as *Saccharomyces cerevisiae* and *Candida albicans*) and mold fungi (such as *Aspergillus niger*).

## 2. Results

### 2.1. Melissopalynological Analysis

On the basis of melissopalynological analysis, 7 varietal honeys and 4 multifloral honeys were distinguished. Among the varietal honeys, the following were distinguished: plum honey (P) with the *Prunus* type (46.98%) dominant pollen; willow honey (Sa) with the *Salix* sp. (70.25%) dominant pollen; rapeseed honey (Br) with the Brassicaceae type (81.70%) dominant pollen; lime honey (Tc) with the *Tilia* sp. (28.99%) dominant pollen; *Phacelia* honey (Ph) with the *Phacelia tanacetifolia* (65.62%) dominant pollen; honeydew honey (So) with the *Solidago* type (46.48%) dominant pollen; and sunflower honey (He) with the *Helianthus* type (73.35%) dominant pollen (Appendix A).

In the case of multifloral honeys, they were characterized as follows: multifloral-Br (MBr) with a predominance of pollen from Brassicaceae type (33.01%), *Aesculus hippocastanum* (15.53%) and *Polygonum bistorta* (15.53%); multifloral-Sa (MSa) with a predominance of pollen from *Salix* sp. (21.55%) *Solidago* type (17.24%) and *Tilia* sp. (17.24%); multifloral-AP (MAP) with a predominance of pollen from *Acer* sp. (37.38%) and *Prunus* type (37.38%); and multifloral-P (MP) with a predominance of pollen from *Prunus* type (29.47%), Brassicaceae type (15.79%) and *Salix* sp. (15.26%) (Appendix A).

### 2.2. Physicochemical Properties of Honey

Table 1 shows physicochemical properties of 11 tested honeys (average ± standard deviation). The water content in the tested honey samples was within the range of 14.67 ± 0.47–18.00%.

Another analyzed parameter was electrical conductivity, which makes it possible to distinguish between nectar and honeydew honeys. The electrical conductivity was within the range of 0.25 (sunflower He)–0.91 mS·cm^−1^ (honeydew So). The results obtained were within the values applicable for nectar honeys, i.e., 0.2 ÷ 0.8 mS·cm^−1^, and for honeydew, i.e., above 0.8 mS·cm^−1^.

The phenolic compounds present in honey come from honeydew or pollen. It was observed that the content of phenolic compounds in dark honeys, e.g., honeydew (So) 808.05 ± 7.20 μg GAEs/g, was higher than the content of these compounds in light honeys, e.g., multifloral (MBr) 404.74 ± 9.12 μg GAEs/g and sunflower (He) 431.27 ± 5.45 μg GAEs/g. Rapeseed honey (Br) 378.27 ± 7.3 μg GAEs/g was characterized by the lowest content of phenolic compounds.

Among the tested honeys, the following honeys had the highest protein content, i.e., above 100 mg/mL: multifloral-AP (116.80 ± 0.57 mg/mL), plum P (112.40 ± 2.47 mg/mL), willow Sa (107.60 ± 0.57 mg/mL) and multifloral-P (103.60 ± 0.57 mg/mL). Sunflower He (41.20 ± 2.47 mg/mL) and rapeseed Br (49.20 ± 3.40 mg/mL) honeys had the lowest protein content below 50 mg/mL.

The Pfund scale includes seven classes of honey colors. Among the tested honeys, the following colors were distinguished: extra-light amber (18.18%), light amber (45.45%) and amber (36.36%) (Appendix A). In our study, sunflower honey was the brightest (He) (white with 0.36 average absorbance) and honey was the darkest (amber with average absorbance 2.94) multifloral-AP (MAP).

Principal component analysis (PCA) of the physicochemical properties of honeys tested synthetically showed their diversity depending on type. The eigenvalues of the first two axes were 2.47 and 1.13. The first axis explained over 49% and second axis over 22% of the variability of the analyzed data/physicochemical properties of studied honeys, and all four axes over 98%. This proves the major role axes 1 and 2 play in ordering the variables and determining the factors responsible for the distribution of honey types in the ordination diagram (Figure 1). All the variables analyzed, except for protein content for axis 2, were statistically significant at the level of *p* < 0.05. The ordination diagram showed two main trends of the variation in the physicochemical properties of the tested honey (Figure 1). The first one was related to the first axis and positively correlated with all variables tested, except the water content. The strongest correlation with this axis was shown by total phenolics, pH and protein content. This axis determined the gradient of the content of the analyzed properties in the honey types. Group I of the studied honeys (right side of the ordination diagram) represents an increasing content of total phenolics, pH, and protein content, starting from honeydew (So), multifloral (MAP), multifloral (MP), plum (P) and willow (Sa) honey. Group II (left side of the PCA diagram) was negatively correlated with the first axis and characterized by a high water content and lower content of the phenolics, proteins and pH values. This group of honeys includes rapeseed (Br), sunflower (He), lime (Tc), multifloral (MBr) and phacelia (Ph). The second axis of the PCA ordination diagram was strongly and positively correlated with electrical conductivity and water content, and the second axis determined the gradient of replaced variables in the studied honeys (Group III). The water content increased from the multifloral (MP) (14.67%) and willow (Sa), multifloral (MSa) and sunflower (He) (located under the second axis and negatively correlated with it) to multifloral (MAP), multifloral (MBr), and lime (Tc), positively correlated with the discussed axis, where the highest content of water was recorded. Honeys exhibiting highest conductivity are: honeydew (So), multifloral (MSa), lime (Tc) and phacelia (Ph) (Figure 1).

Sugars are main components of honey (Table 2). In order not to miss any of the sugars, we used 16 sugar standards, including mono-, di- and trisaccharides. In Table 2 we presented only the sugars that were determined by the HPLC analysis. A representative HPLC profile of honey number 11 is shown in Appendix A. Simple sugars, i.e., glucose and fructose, were identified in the highest amount in all the tested honey samples.

The PCA ordination analysis shows relationships between the honey type and diversity of sugars and their content (Figure 2). All analyzed sugar types were statistically significant at the level of *p* < 0.05 for the two first axes of the ordination PCA diagram, except glucose for axis 2. The first axis explains ca. 53% (eigenvalue 3.18) and the second axis ca. 32% (eigenvalue 1.9) of the data variability. All four axes explain over 97% of the data variability.

Axis 1 is positively correlated with all the variables tested, except the glucose content. The first axis determines the falling share of glucose in the honey types form the right side of the PCA diagram for sunflower (He), willow (Sa), multifloral (MP) and plum (P) to multifloral (MAP) in the middle, and lime (Tc), phacelia (Ph) on the left side. In relation to the content of other sugars, axis 1 is positively correlated with them. The strongest correlation with sucrose and rhamnose is observed and depicted by the axis showing a rising gradient of these sugars from rapeseed (Br), willow (Sa) and multifloral (MSa) to lime (Tc) and phacelia (Ph).

Axis 2 of the diagram is strongly, positively correlated with the erlose content and strongly negatively correlated with the fructose and fucose content. The erlose content decreases form rapeseed Br (8.26 g/100 g), plum P (3.02 g/100 g) and willow Sa (2.92 g/100 g) through honeydew So, multifloral MSa, multifloral MBr and multifloral MAP honey, where the erlose content ranges from 1.77 g/100 g to 0.66 g/100 g respectively to lime (Tc) and phacelia (Ph) honey in which no erlose was found. The high fucose content in sunflower (He) and multifloral (MAP) honey is positively correlated with fructose.

The ratio of fructose to glucose was typical for honey. The more glucose a honey has, the faster it tends to crystallize. In honey, the ratio of fructose to glucose should ideally range from 0.9 to 1.35. A fructose to glucose ratio below 1.0 leads to faster honey crystallization, whereas crystallization becomes slower when this ratio is more than 1.0 [18,19,20]. In the present study, the average ratio of fructose to glucose was around 1. However, two tested honeys (Tc and Ph) had a ratio well below 1.0 (0.85 and 0.71, respectively), which indicates greater chances of honey crystallization (Table 2).

### 2.3. Antimicrobial Activity of Honey

The antimicrobial activity of the honey samples was expressed by the inhibition of the growth of the tested bacteria around the wells on the agar medium, and it varied (Appendix A). The Gram-positive bacteria *B. circulans* proved to be the most sensitive to the activity of the honeys. The inhibition zones of bacterial growth were observed in all concentrations (62.5–500 mg/mL) in seven honey samples: plum (P), rapeseed (Br), lime (Tc) and multifloral (MBr, MAP, MP, MSa). In the case of three honeys—willow (Sa), phacelia (Ph), and sunflower (He)—no activity against *B. circulans* was found at concentrations of 125 and 62.5 mg/mL. On the other hand, honeydew honey (So) did not inhibit bacterial growth only in the concentration of 62.5 mg/mL. Taking into account the antibacterial activity observed after the use of the lowest concentration of honeys (62.5 mg/mL), it should be stated that the following honeys proved most effective against *B. circulans*: multifloral (MSa), plum (P) and rapeseed (Br) (growth inhibition zones of 15.22, 14.17, 13.55 mm respectively).

The results of analysis of the significance of difference test showed that the type of honey and its concentration are the factors influencing the antibacterial activity of honeys against *B. circulans*. The highest activity, expressed by the size of the inhibition zone, was observed for rapeseed honey (Br). This result is significantly different from plum (P) and multifloral (MAP) honey, which show similar activity, and willow (Sa), phacelia (Ph) and sunflower (He) (Figure 3). At a lower concentration (Figure 3), rapeseed (Br), lime (Tc), multifloral (MSa), plum (P) and multifloral (MBr) are less efficient, but retain their antibacterial properties, which significantly differs from willow honeys (Sa), phacelia (Ph) and sunflower (He), which show no such activity. At the lowest concentration, rapeseed (Br), multifloral (MSa), plum (P) and smaller multifloral (MBr) honeys show high activity, which significantly differs from the others, which have lost their properties (Figure 3).

For all tested honey types, inhibition of bacterial growth in all tested microorganisms at the highest concentration of 500 mg/mL was visible (Figure 4, Figure 5 and Figure 6).

It should be noted that the tested honey varieties showed significantly lower activity against other Gram-positive bacteria used in the experiments, i.e., *S. aureus*. In this case, the inhibition zones of bacterial growth were observed only after the application of 50% honey concentrations. The growth of *S. aureus* was most strongly inhibited by these honeys: multifloral MBr (8.42 mm) and multifloral MSa (9.97 mm) (Figure 4).

Similarly, only at the concentration of 500 mg/mL did the tested honeys inhibit the growth of Gram-negative bacteria *E. coli*. The largest zones of growth inhibition (9.6–11.9 mm), and thus the highest activity, were recorded for the following honeys: multifloral MBR and MAP. The exception was the sunflower honey (He), which showed no activity against this bacteria (Figure 5).

At a lower concentration (250, 125, 62.5 mg/mL), no tested honeys caused a decrease in *E. coli* and *Staphylococcus aureus* growth. A broader effect was evident when testing was done with honey at lower concentrations in relation to *B. circulans* and *A. niger* (Figure 3 and Figure 7). In addition, based on the results obtained by the diffusion method, it was found that *P. aeruginosa* bacteria, both standard and clinical strains, was the microorganism completely insensitive to the honey varieties being tested.

The analyzed honeys show inhibitory activity against *E. coli* and *S. aureus* only in the highest concentration (Figure 5 and Figure 6). Multifloral MAP, MP, and MBr honeys have the highest activity against *E. coli* and statistically significantly differ in this respect from multifloral (MSa) and plum (P) (Figure 6) honeys. However, the activity of honeys multifloral (MSa) and (MBr) against *S. aureus* is statistically significantly different from the properties of multifloral honeys MAP and MP and lime (Tc) (Figure 6).

### 2.4. Antifungal Activity of Honey

The antifungal activity of the honey samples used at concentrations ranging from 62.5 to 500 mg/mL was tested against *A. niger*, *C. albicans* and *S. cerevisiae* using the radial diffusion method. On the basis of the obtained results, it was found that *C. albicans* and *S. cerevisiae* showed resistance to the tested honey samples at all concentrations.

On the other hand, the tested honey varieties effectively inhibited the growth of *A. niger* (Appendix A). The maximum antifungal activity was found in all honey samples at the concentration of 500 mg/mL in the range from 62 to 99.25 µg/mL based on the properties of amphotericin B (µg/mL). At this concentration, the most active honeys were multifloral (MSa), plum (P) and honeydew (So), which differed significantly from multifloral (MBr) and phacelia (Ph) honeys (Figure 7). At a lower concentration (Figure 7), the properties of multifloral (MP), multifloral (MSa) and willow (Sa) honeys are comparable and significantly different from phacelia (Ph). At the next concentration, i.e., 125 mg/mL (Figure 7), multifloral (Msa) and honeydew (So) honeys retained antifungal properties, being significantly different from rapeseed (Br) and phacelia (Ph). At the lowest concentration (Figure 7) willow (Sa) and multifloral (MBr) honeys were most active, differing from plum (P) and rapeseed (Br), which showed lowest antifungal activity.

### 2.5. Catalase

All the honey samples with catalase addition had the same or similar growth inhibition zones compared to the control, i.e., honey without catalase. The tested honeys remained active against *B. circulans*, *E. coli* and *S. aureus*, which proves that the activity was related to other factors and that hydrogen peroxide did not affect the antimicrobial activity of these honeys (Appendix A). Positive control data with different dilutions of hydrogen peroxide are presented in Appendix A.

### 2.6. Lysozyme-like Activity of Honey

In subsequent experiments, lysozyme-like activity was checked by applying the tested honey samples to plates containing *M. lysodeikticus* according to the procedure of Mohrig and Messner [10]. Lysozyme-like activity was found in all the tested honeys. The highest lysozyme-like activity corresponding to the activity of 447.26 ug/mL and 159.74 ug/mL EWL was measured in multifloral honeys MAP and MP. The other varietal honeys have low lysozyme-like activity. Comparable values were obtained for the following honeys multifloral (MSa), willow (Sa), multifloral (MBr), sunflower (He) and plum (P), rapeseed (Br), lime (Tc), phacelia (Ph), honeydew (So), which were statistically significantly different from other samples (Figure 8).

Lysozyme-like activity level was tested in all samples taken at various steps of honey preparation, i.e., after centrifugation, dialysis and lyophilization (Table 3). The peptidoglycan digestion zone is shown in Appendix A. The highest lysozyme-like activity was observed in honey after centrifugation (2.3 ± 0.47 µg/ml EWL).

Our results showed that there was activity against *M. lysodeikticus* at each step in the honey samples, which is defined as lysozyme-like activity. In order to find out whether there is any lysozyme protein in honey, it is necessary to perform other long-term experiments.

### 2.7. HPLC Analysis of Phenolic Compounds in Honey Samples

The findings are presented in Table 4 and Figure 9. The presence of caffeic and syringic acid in various amounts was found in all tested honeys. Some honeys identified coumaric acid (in 45% of samples) and cinnamic acid (in 73% of samples). The highest content of caffeic acid was observed in the following honeys: phacelia (Ph)—356.72 µg/g, multifloral Sa (MSa) and multifloral Br (MBr)—318.9 µg/g, and cinnamic acid in willow honey (Sa)—11.9 µg/g. The content of coumaric and syringic acid in the honey samples did not exceed 10 µg/g.

## 3. Discussion

Due to a widespread occurrence of multidrug-resistant (MDR) bacterial and fungal strains, there is an urgent need to look for antimicrobial substances. Nosocomial infections make up a very high percentage of postoperative complications and are very difficult to treat. Therefore, honey with its antimicrobial properties is a very promising substance with many valuable properties [21]. In the honeys tested in our study, similarly to earlier publications [22,23,24,25,26], several substances with antimicrobial properties were identified. Although honey has some limitations and cannot be used as a drug, it can still enhance drug treatment against MDR bacterial and fungal strains.

In honey, phenolic acids are one of the most important groups of compounds with antimicrobial activity. Phenolic acids and flavonoids were recognized in the 1990s as important antibacterial substances. In studies of various honeys from Burkina Faso, it was found that honeydew honeys had the highest content of phenolic compounds 113.05 ± 1.10–114.75 ± 1.30 mg GAE/100 g [27]. Moreover, the level of phenolic compound profile can be the marker for authentication of the botanical and geographic origin of the honey [25,28]. The composition of phenolic compounds depends on plant source. Therefore, in this study, the total phenol content was measured and the phenolic compound profile outlined using some of the most important typical compounds was illustrated. The syringic acid and vanillic acid as an example of compounds belonging to general group of benzoic acids and caffeic acid, *p*-coumaric acid as an example of compounds belonging to general group of hydroxycinnamic acids were selected. In our study, the highest amount of phenolic acids was found in the honeydew honey (808.05 µg GAE/g, Table 1 and Table 4, Figure 9) with the highest antifungal properties aiming at *A. niger* (Figure 7). Among the tested phenolic acids, caffeic acid was the most abundant, which was found in the highest amounts in the following honeys: phacelia (Ph)—356.72 µg/g, multifloral (MSa) and multifloral (MBr)—318.9 µg/g (Table 1). The highest bactericidal activity against *S. aures* was found in multifloral honeys MSa and MBr. Moreover, multifloral MSa honey at all concentrations showed high antifungal activity (*A. niger*). Additionally, the highest amounts of syringic acid and cinnamic acid were identified in rapeseed honey (Br) (Table 4). In a study by Chong et al. [29], it was shown that caffeic and syringic acid had antibacterial and antifungal activity. In addition, caffeic acid was bactericidal against *S. aureus* [30]. On the other hand, cinnamic acid shows antifungal properties against *A. niger*, *C. albicans* and antibacterial, among others, against *Mycobacterium tuberculosis* and *E. coli* [14,31]. The abovementioned compounds are connected with the antimicrobial effect of the most effective honeys tested in our study. At the highest concentration (500 mg/mL), multifloral honey (MAP) showed the highest bactericidal activity against *E. coli* (inhibition zone: 11.9 mm), and multifloral honey (MSa) against *S. aureus* (inhibition zone: 9.9 mm). Additionally, multifloral honey (MBr) is effective against both bacteria: *E. coli* (inhibition zone: 9.6 mm) and *S. aureus* (inhibition zone: 8.4 mm) (Figure 4, Figure 5 and Figure 6). The antimicrobial properties against bacteria and fungi are agreed to exist if the inhibition of zone is greater than 6 mm [32,33]. The honeys tested in our study also showed antifungal activity, e.g., on *A. niger*. However, there was no fungicidal activity against *C. albicans* and *S. cerevisiae*. The highest activity against *A. niger* was observed in multiflower (MSa) and honeydew (So) honeys (Figure 7 and Appendix A). Most likely, due to the high content of phenolic compounds, multiflorous honeys had this high antifungal activity [34]. Furthermore, polyphenolic compounds can interact with other active molecules present in honey and their synergistic effect may be responsible for antibacterial activity of different honeys [35]. The activity against various bacteria, including *Bacillus cereus*, *S. aureus* and *E. coli*, was tested in a 75% and 50% solution of multifloral honey from Turkey [36]. The results indicated that at a higher concentration, multifloral honey showed bactericidal activity against *S. aureus* (inhibition zone: 0–7 mm) and *B. cereus* (inhibition zone: 0–6 mm). No activity was demonstrated in either concentration against *E. coli* [37]. In multifloral honeys from Spain, the activity against *S. aureus* was checked by the method of agar well diffusion in a 75% honey solution. Osés et al. [36] found that the tested honeys showed an inhibitory effect on *S. aureus* in the form of inhibition zones of bacterial growth 14.05 ± 2.31 mm. An experiment by Alvarez-Suarez et al. [38] tested the activity against *S. aureus* of multiflower honey from Cuba produced by two species of bees: *Melipona beecheii* and *Apis mellifera*. The authors found that honey produced by *M. beeicheii* showed about sevenfold the activity against this bacteria of honey produced by *A. mellifera* [38,39]. Honey in various concentrations (10%, 20%, 30%, and 100%) from Pakistan showed different degrees of activity against *A. niger* and *Penicillum chrysogenum* [34]. Moussa et al. [40] showed no activity against *C. albicans* honey from Algeria. By contrast, Irish et al. [41] found that different honeys inhibit clinical isolates of *C. albicans*, *C. glabrata* and *C. dubliniensis*. Hence, honey is important in combating fungal infections that arise in immunocompromised patients, which may lead to the development of opportunistic infections [5].

Osmosis is an important physical phenomenon connected with antimicrobial properties of honey. High sugar content exerts osmotic pressure on bacterial cells, which results in water loss in bacterial cells. Dehydrated cells are unable to grow and develop in hypertonic sugar solution [21,25,42]. Furthermore, osmotic pressure can affect the ability of bacteria to form biofilms [43]. The presence of sugars in honey can also interfere with bacterial quorum sensing [21]. Wahdan et al. [28] showed that fungi are more tolerant to osmosis compared to bacteria and the sugar solution did not inhibit the growth of *C. albicans* [39]. Low water content inhibits yeast fermentation and bacterial growth [26]. The composition of the honeys tested in our study consists mainly of sugars and water, and also in smaller amounts phenolic compounds and proteins. Water content in the tested honeys was within the normal ranges accepted for honeys according to the International Honey Commission [44], i.e., from 14.6 to 18.0% (Table 1, Figure 1). The highest content of glucose was recorded in phacelia honey (Ph)—53 ± 0.46 g/100g, while the highest content of fructose was found in multifloral honey (MAP)—43.57 ± 0.28 g/100g (Table 2). Moreover, rhamnose sugar was detected in the highest amount in multifloral honey (MAP), which showed the greatest activity against *E. coli.* Erlose was a characteristic sugar found in these honeys, which is formed by the action of invertase on sucrose. The presence of erlose in honey was first confirmed by White and Maher in 1953 [45]. Erlose is an intermediate trisaccharide in the metabolism of nectar sugars by honeybees [46]. The highest erlose content (8.26 g/100 g, Table 2) was recorded in rapeseed honey (Br), which showed the highest activity against *B. circulans*. Additionally, this honey had the highest sucrose content (4.96 g/100 g, Table 2) among the tested honeys. Also, in rapeseed honey from various regions of Poland, the presence of sucrose was identified: 0.5–2.4 g/100 g [47]. On the other hand, in rapeseed honey from Germany, no sucrose or erlose was detected [48].

Another important physical factor that affects the antimicrobial activity of honey is pH. Low pH ranging from 4.08 (lime honey—Tc) to 4.96 (willow honey—Sa) was observed in our study (Table 1). The low pH in honey is due to the presence of organic acids in honey, which include gluconic acid with antimicrobial activity formed by the oxidation reaction of glucose by glucose oxidase [22,49].

In our study, we showed multifloral honey to work best against *E. coli* bacteria (MAP). It is also characterized by the highest content of proteins (116.80 mg/mL, Table 1) and lysozyme-like activity (447.26 µg/mL EWL, Figure 8) among all tested honeys. Lysozyme is active against Gram-positive bacteria by acting on peptidoglycan. Gram-negative bacteria, e.g., *E. coli* are not susceptible to the action of lysozyme due to the presence of the outer membrane. Based on the morphological and immunocytochemical studies by Wild et al. [50], it has been illustrated that lysozyme does not act on membranes but on *E. coli* cytoplasm, leading to its degradation. In order to clarify the action of lysozyme on *E. coli*, Wild et al. [50] additionally used cryotechnics. They found that lysozyme can bind to the outer membrane and penetrate the periplasmic space, possibly reaching the inner cell membrane. Moreover, Wild et al. [50] conducted antimicrobial tests which showed that lysozyme is bactericidal against *E. coli*, but does not completely break down the bacteria. Two years later, Pellegrini et al. [51] showed that lysozyme inhibits DNA and RNA synthesis. In addition, it has been found that lysozyme causes damage to the outer cell membrane and permeabilization of the inner membrane, which results in the death of *E. coli* bacteria. In contrast, ultrastructural studies showed no effect of lysozyme on bacterial morphology [51]. The mechanism of the bactericidal activity of lysozyme on Gram-negative bacteria requires further research.

There is an urgent need for new substances with antimicrobial capabilities against which pathogenic bacteria and fungi do not develop resistance [1,2,52,53]. Novel varietal honeys tested in our study show a broad spectrum of antibacterial and antifungal activities. This may suggest that the studied honeys may act as natural products that could reduce the effects of fungal and bacterial infections. Compounds in honeys, such as lysozyme-like and phenolic acids, i.e., coumaric, caffeic, cinnamic and syringic acids, played a key role in the health-benefit properties of honeys tested in our study. Furthermore, as in other studies, polyphenolic compounds can interact with other active molecules present in honey, and their synergistic effect may be responsible for antibacterial activity of different honeys [35].

## 4. Materials and Methods

### 4.1. Honey Sample Collection and Classification

The experiments were carried out with 11 honey samples originating in Poland, collected in 2018 and grouped in Table 1. The honeys were classified according to the standard methods recommended by the European Union [54]. Then, the honeys were grouped in terms of the dominant pollen or most common pollen in the honey sample (Appendix A). The flowering periods of plants from which the pollens originated, were given after the Biolflor Database (Trait Database of the German Flora: http://www.ufz.de/biolflor accessed on 1 July 2022) and the possibility to collect given variety of honey [55,56].

### 4.2. Honey Sample Classification Using Pollen Analysis

Ten rams were weighed from each honey sample, 20 mL of distilled water was poured into them and then heated on a water bath until the honey samples completely dissolved. The obtained solution was subjected to centrifugation in an MPW 341 centrifuge with a horizontal rotor at a speed of 3000 rpm (MPW Med. Instruments, Warsaw, Poland). Next, the liquid was decanted, but about 5 mL of suspension was left. The solution was poured into smaller test tubes and centrifuged again, maintaining the previous parameters. The liquid was then decanted again, leaving 2 mL of suspension above the sediment of pollen grains. Fifty microliters of the suspension was taken and applied to microscope slides. Two preparations were made of each honey sample. The microscopic analysis was carried out with the Olympus CX21 microscope (600×) (Olympus, Shinjuku, Tokyo, Japan). An average of 300 pollen grains of nectariferous plants were counted and classified to the lowest possible taxon.

Pollen grains were classified into dominant pollen ≥ 45%, accompanying pollen between 16% and 45%, single pollen between 3% and 16%, and occasional pollen ≤ 3%. If the share of the leading taxa were more than or equal to 45%, such honey was classified as nectar-varietal honey.

### 4.3. Honey Sample Preparation

Two grams of each honey sample was weighed in sterile beakers and dissolved in 2 mL of sterile water. Samples prepared in such a manner were incubated at 37 °C for about 3 h in the incubator, stirring several times until the honey was dissolved completely. Immediately before use, honey samples were twice diluted with sterile water to obtain the following dilutions: 1:2 (500 mg/mL), 1:4 (250 mg/mL) 1:8 (125 mg/mL) 1:16 (62.5 mg/mL) which were used in further analyses.

### 4.4. Physicochemical Properties of Honeys

#### 4.4.1. Water Content

The water content of honey was checked with the PAL-22S refractometer (Conbest, Cracov, Poland). Each honey sample was thoroughly mixed and a drop of liquid honey was transferred to the prism of a refractometer according to the manufacturer instruction. Each honey sample was checked in triplicate.

#### 4.4.2. Electrical Conductivity

The electrical conductivity in honey was measured with the CC-105 electrical conductivity meter (Elmetron, Zabrze, Poland) at 20 °C. Twenty grams of honey was dissolved in 100 mL of distilled water and in such solution the electrical conductivity of honey sample was measured. Each honey sample was checked in triplicate [44].

#### 4.4.3. pH

The pH of honey sample was measured in a 10% honey solution using an analogue pH meter (HANNA Instruments, Olsztyn, Poland). Each honey sample was checked in triplicate.

#### 4.4.4. Color Intensity

Honey color was determined using the Pfund scale according to the USDA (United States Department of Agriculture, United States Standards for Grades of Extracted Honey) classification [55]. Pure honey samples were heated at 60 °C in a water bath until their complete dissolution. Next, samples were placed in 10 mm cuvettes and the absorbance (λ = 560 nm) was measured, using deionized water as a blank. The absorbance results were multiplied by a 3.15 factor. The obtained results were compared to the values presented in Appendix A after [56] and the color of tested honeys was determined and presented in Appendix A.

#### 4.4.5. Total Phenolic Content

The content of phenolic compounds was determined with a spectrophotometric method using the Folin–Ciocâlteu reagent (Sigma-Aldrich, Saint Louis, MO, USA) [57,58]. One gram of honey sample was dissolved in 20 mL of distilled water. Five mL of 0.2N Folin-Ciocalteu reagent was added to 1 mL of honey solution. Then, after a 5 min incubation, 4 mL of 75% *w*/*v* aqueous sodium carbonate solution was added to the solution and incubated for 2 h at room temperature. After this time, the absorbance was measured (λ = 765 nm), using a distilled water as a blank. The total phenolic content was calculated on the basis of a standard curve prepared for known concentrations of gallic acid (5–100 μg/mL) (Sigma, EC 3.2.1.17) and was expressed in μg of gallic acid equivalent (GAE) per g of honey. All analyses were made in triplicate.

#### 4.4.6. Sugar Analysis in Honey Samples

Sugar profiles of 11 honey samples were analyzed by HPLC using the Shimadzu chromatographic system (Kyoto, Japan) with the RID-10A refractive index detector. The mobile phase (Milli-Q water obtained using the Elix^®^ Essential 3 Water Purification System with Synergy^®^ UV Water Purification System, Merck Millipore, Darmstadt, Germany) was run at a flow rate of 0.6 mL/min at 75 °C through the REZEX RPM-Monosaccharide Pb^2+^ column (300 × 7.8 mm, Phenomenex, Torrence, USA). The column was calibrated using sixteen carbohydrate standards, including mono-, di- and trisaccharides. Standard solutions of mono-, di- and trisaccharides: glucose, fructose, galactose, rhamnose, xylose, mannose, sucrose, turanose, maltose, celobiose, fucose, trehalose, melibiose, erlose, melezitose and raffinose (Sigma-Aldrich, Saint Louis, MO, USA) were used for interpretation and quantification of sugars in the honey samples. Sugar concentrations were expressed in g/100 g honey.

#### 4.4.7. Protein Content

The protein content was determined by the Bradford method [59] in 50% (*w*/*v*) honey samples solutions. Twenty microliters of such a solution was added to 1 mL of Bradford’s reagent (Bio-Rad, Hercules, CA, USA) (Coomassie Brilliant Blue G-250), using deionized water as a control. After 5 min of incubation, absorbance was measured at 595 nm using bovine serum albumin in deionized water as a standard (0.1–0.9 mg/1 mL).

### 4.5. Microorganisms Used in the Antimicrobial Assays

The antimicrobial activity of honey samples was tested against the following bacteria:*Escherichia coli* D31 (CGSC 5165; Genetic Stock Centre, New Haven, CT, USA)*Bacillus circulans* strain ATCC 61;*Staphylococcus aureus*, clinical strain 1-KI, obtained from The Department of Immunobiology, Maria Curie-Sklodowska University in Lublin, Poland.*Pseudomonas aeruginosa* strain ATCC 27853;*Pseudomonas aeruginosa* clinical strain 02/18, obtained from The Department of Microbiology and Epidemiology, Military Institute of Hygiene and Epidemiology in Warsaw, Poland.

The bacteria were cultured in Luria-Bertani broth (LB; Biocorp, Warsaw, Poland) at 37 °C for 24 h.

The following fungi strains were obtained from the fungal collection of the Department of Immunobiology, UMCS Lublin (Poland):*Aspergillus niger* 71, grown in PDA broth (5% potato extract; 0.5% dextrose; 1.7% agar);*Saccharomyces cerevisiae*, subcultured in sterile Sabouraud broth (1% pepton; 4% glucose; 1.5% agar);*Candida albicans*, grown in YPD broth (1% yeast extract; 2% pepton; 2% glucose; 1.6% agar).

### 4.6. Antibacterial Activity Assay

The presence of antibacterial activity in honey samples was detected by the radial diffusion assay on solid agar plates containing appropriate bacterium (150 μL) in the amount of 1.5–4.2 × 10^6^. Each well on the Petri plates was filled with 5 μL of honey samples (1:2–1:16), next the agar plates were incubated for 24 h at 37 °C. The diameters of bacteria growth inhibition zones were measured with digital caliper (Pro, Bielsko-Biała, Poland) and expressed in millimeters. The experiment was repeated four times.

### 4.7. Antifungal Activity Assay

Antifungal activity was detected by a diffusion well assay against *A. niger* using PDA plates (8 mL) containing about 1.6 × 10^6^ spores/mL of the medium. Each well on the petri plates was filled with 5 μL dilutions of honey (1:2–1:16). Agar plates with PDA medium were incubated for 24 h at 28 °C, next the diameters of *A. niger* growth inhibition zones were measured with a digital caliper (Pro, Bielsko-Biała, Poland). The obtained results in millimeters were calculated on equivalent of amphotericin B (μL/mL).

In the case of *C. albicans* and *S. cerevisiae*, 24 h fungi culture was standardized to 0.5 McFarland. After incubation, the reaction mixture was sown in the amount of 100 μL on agar plates (1.6%) with Sabouraud medium (10 mL). The appropriate dilutions of honey (1:2–1:16) were added to the wells in the medium. The plates were incubated at 37 °C for 24 h. After incubation the diameters of growth inhibition zones in millimeters were measured with a digital caliper (Pro, Bielsko-Biała, Poland).

### 4.8. Lysozyme-like Activity of Honey Samples

Lysozyme-like activity of honey samples was checked using agarose plates containing freeze-dried *Micrococcus lysodeikticus* (Sigma-Aldrich, Saint Louis, MO, USA) [10].

The activity was tested in the following steps of preparation of the honey samples: in the mixture after overnight shaking and centrifuged (sample 1); in the supernatant after dialysis (sample 2); in the lyophilized samples (sample 3) (Appendix A).

Each well on the petri plates was filled with 5 μL samples, and next plates were incubated at 28 °C for 24 h. After this time, peptidoglycan digestion zones were measured. The lysozyme-like activity was defined as an equivalent of EWL activity (μg/mL) (Sigma, EC 3.2.1.17). Similarly, for control plates, wells were filled with egg-white lysozyme (EWL) (Appendix A). The level of lysozyme-like activity was calculated on the basis of a standard curve prepared for known concentrations of lysozyme EWL (μg/mL) (Sigma-Aldrich, Saint Louis, MO, USA).

Additionally, lysozyme-like activity level was tested in all samples taken at various steps of honey preparation, i.e., after centrifugation, dialysis and lyophilization (Appendix A). Peptidoglycan digestion zone is shown in Appendix A. The level of lysozyme-like activity was calculated on the basis of standard curve prepared for known concentrations of lysozyme EWL (μg/mL) (Sigma-Aldrich, Saint Louis, MO, USA) (Appendix A).

### 4.9. Antimicrobial Activity Connected with Hydrogen Peroxide in Honey Samples

Agar plates (0.7%) with the LB medium (10 mL) (LB; Biocorp, Warszawa, Poland) containing appropriate bacterium (150 μL) in the amount of 1.5–4.2 × 10^6^ were used to detect the antimicrobial activity connected with hydrogen peroxide in honey samples. Each well on the petri plates was filled with 5 μL samples containing appropriate honey dilutions as a control and 5 μL samples containing appropriate honey dilutions with catalase (the enzyme degrading hydrogen peroxide) (Sigma-Aldrich, Saint Louis, MO, USA), as a test samples (Appendix A). Next, plates were incubated at 37 °C for 24 h and the diameters of bacterial growth inhibition zones were measured with a digital caliper (Pro, Bielsko-Biała, Poland).

Positive control samples: Each well on the petri plates was filled with 5 μL of freshly diluted 10%, 5%, 3%, and 1.5% hydrogen peroxide (Chempur, H_2_O_2_ -34.01 g/mol 30% pure p.a. CAS: 7722-84-1) diluted in sterile water or honey (number 11), next the agar plates were incubated for 24 h at 37 °C. The diameters of bacteria growth inhibition zones were measured with digital caliper (Pro, Bielsko-Biała, Poland) and expressed in millimeters. The experiment was repeated three times.

### 4.10. Solid Phase Extraction of Honey Samples

Honey samples (5 g) were mixed with 20 mL of deionized water adjusted to pH 2 with HCl and stirred in a magnetic stirrer for 15 min. The samples were then filtered to remove the solid particles. Extraction of phenolic compounds was performed with the Visiprep™ SPE Vacuum Manifold (Sigma-Aldrich, Saint Louis, MO, USA). The SPE cartridges used were Strata-X (500 mg) obtained from Phenomenex (Warsaw, Poland). They were conditioned by washing with 15 mL of methanol, and 20 mL acidified water. Afterwards the filtrated honey sample was passed through a cartridge, which was then washed with 20 mL of deionized water to remove all sugars and other polar constituents of honey. The adsorbed compounds were eluted with 5 mL methanol [60].

### 4.11. HPLC Analysis of Phenolic Compounds in Honey Samples

The concentration of phenolic compounds was quantified by high performance liquid chromatography (HPLC, Agilent Infinity 1260 equipped with DAD detector) (Agilent Technologies, Santa Clara, CA, USA). The HPLC system fitted with Zorbax Eclipse Plus C18 column (100 mm × 4.6 mm × 3.5 µm, Agilent Technologies, Santa Clara, USA) was operated at 40 °C and the flow rate of 1 mL/min. Each 1 µL sample was injected using an autosampler. The mobile phase consisted of 50 mM formate buffer adjusted to pH 4.1 using 1 M NaOH (eluent A) and methanol (eluent B). The elution included an isocratic step with 20% *v*/*v* of eluent B for 1 min after injection of the sample; afterwards, a gradient step of elution (10 min) was applied in the range of 20–90% of eluent B. The separation was ended within 3 min of isocratic elution with 90% of eluent B. The total run time of each analysis was 14 min. After each analysis, a 4 min post run was conducted with 20% of eluent B to restore the start conditions of the analysis.

The components peaks were identified by comparison of retention times of the commercially available standards of the following phenolic acids: *p*-coumaric, caffeic, syringic, vanillic and cinnamic acids. Detection was performed at 280 nm. Agilent OpenLAB CDS ChemStation LC and Ce Drivers (A.02.10 (026) version) software were used for data processing and reporting.

### 4.12. Statistical Analysis

Normal distribution of variables was tested with Shapiro–Wilk tests, and given that not all continuous variables were normally distributed, Kruskal–Wallis H tests (one-way ANOVA on ranks) were performed to compare the mean and standard deviations according to the inhibition abilities of various honey types in four concentrations against selected bacteria and fungi. The results were considered significant at *p* < 0.05. Statistical differences are marked with different letters and their significance at *p* ≤ 0.001 with capital letters, *p* ≤ 0.05 with lower case letters. Statistical analyses were performed using the Statistica 13.2 PL package. To analyze the relationships between species richness inhibition honey activity and physicochemical parameters of honey samples and sugar content, we used multivariate ordination methods in CANOCO version 5.0 package [61,62]. According to the length of the gradient from a preliminary detrended canonical analysis (DCA), a linear model, the principal component analysis (PCA) was used. In the PCA, honey samples were entered as cases and physicochemical parameters and sugar content as dependent variables.

## Figures and Tables

**Figure 1 ijerph-20-02458-f001:**
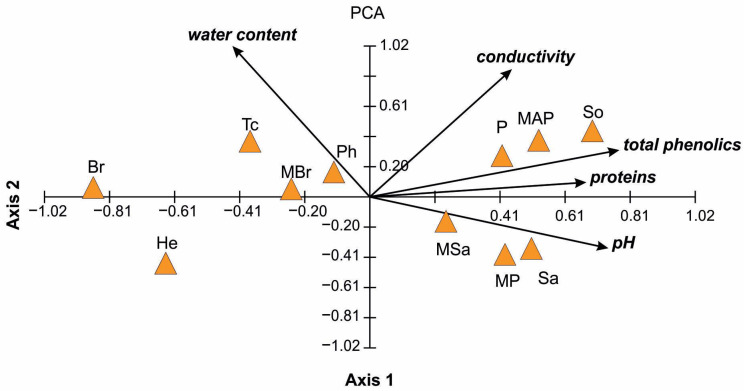
Principal component analysis (PCA) ordination diagram illustrating physicochemical differences among the 11 studied honeys based on five variables (solid line vectors). The variables: total phenolics and pH determined the gradient of axis 1, while the water content of axis 2.

**Figure 2 ijerph-20-02458-f002:**
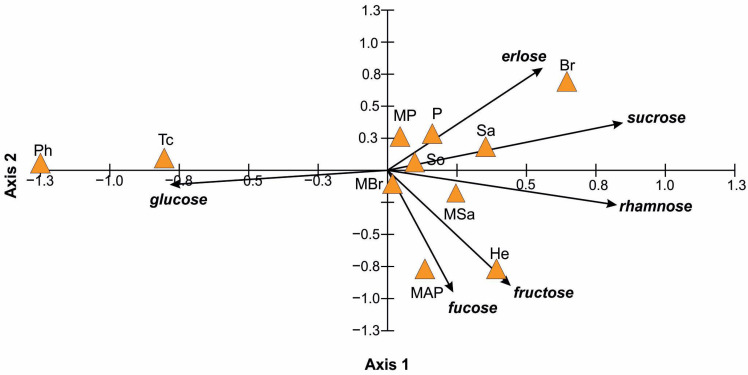
Principal component analysis (PCA) ordination diagram illustrating differences in sugar content among the 11 study honeys (solid line vectors). Content of sucrose determined the positive gradient and glucose content negative gradient of axis 1. In the case of axis 2, the positive gradient was determined by the erlose content and negative by fructose.

**Figure 3 ijerph-20-02458-f003:**
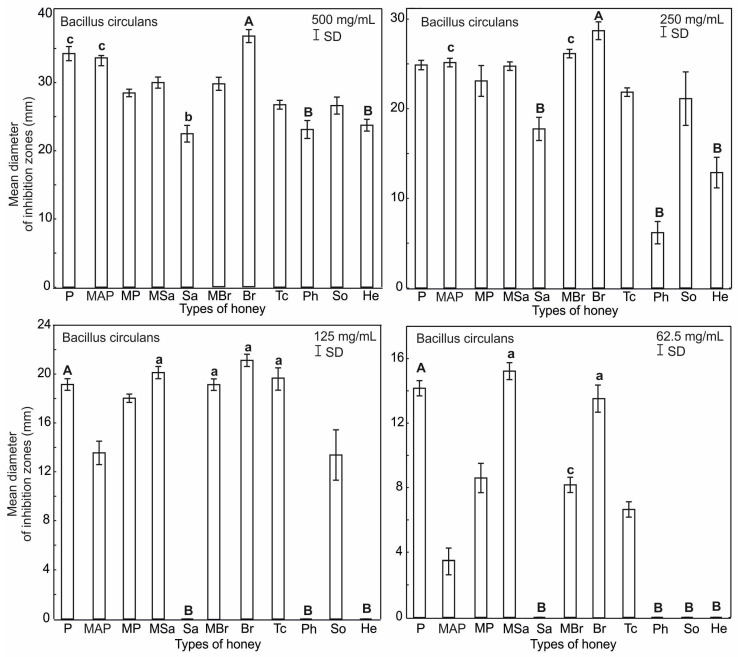
Results of the ANOVA Kruskal–Wallis H test for the mean inhibitory activity of all the tested honeys types and their concentrations against *B. circulans*. Statistical differences are indicated by different letters and their significance at *p* ≤ 0.001 with capital letters (A), *p* ≤ 0.05 with small letters (a). Upper and lower case letters above the bars indicate different levels of statistical significance.

**Figure 4 ijerph-20-02458-f004:**
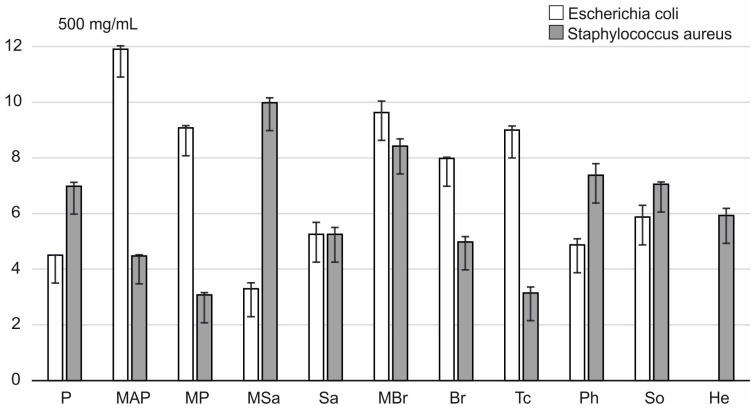
Mean inhibitory activity of tested honey types relative to *E. coli* and *S. aureus*.

**Figure 5 ijerph-20-02458-f005:**
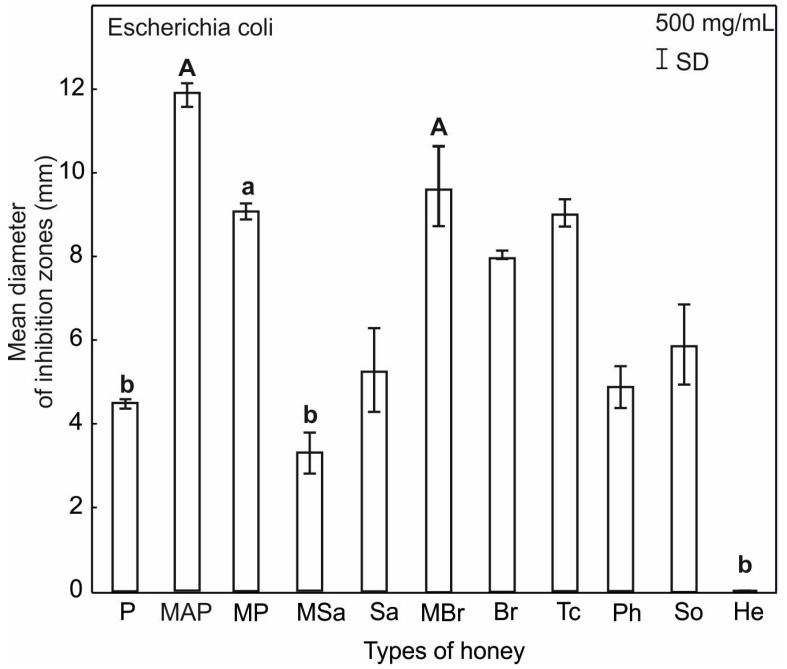
Results of ANOVA Kruskal–Wallis H test for the mean inhibitory activity of all tested honey types against *E. coli*. Statistical differences are indicated by different letters and their significance at *p* ≤ 0.001 with capital letters (A), *p* ≤ 0.05 with small letters (a). Upper and lower case letters above the bars indicate different levels of statistical significance.

**Figure 6 ijerph-20-02458-f006:**
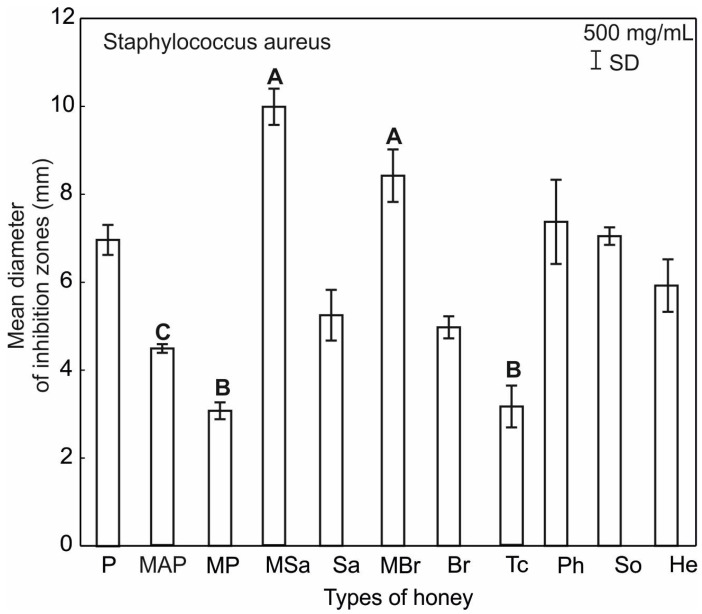
Results of the ANOVA Kruskal–Wallis H test for the mean inhibitory activity of all tested honey types against *S. aureus*. Statistical differences are indicated by different letters and their significance at *p* ≤ 0.001.

**Figure 7 ijerph-20-02458-f007:**
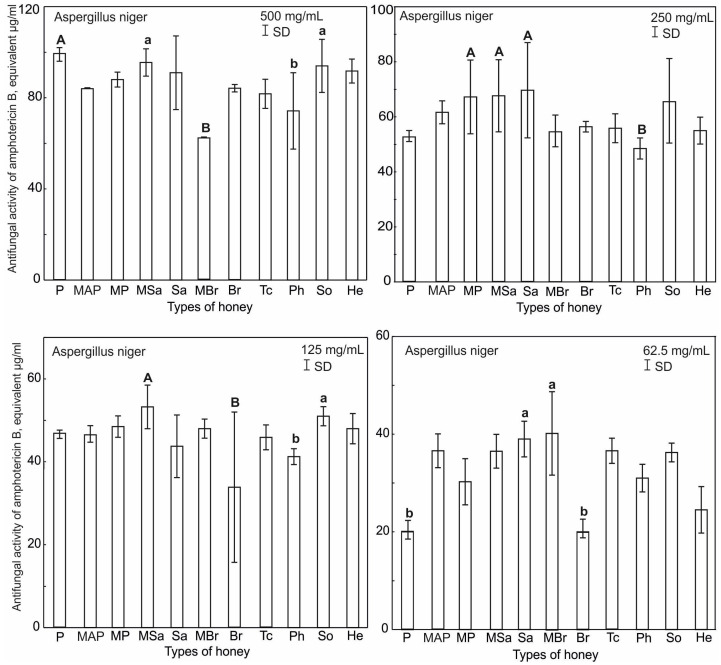
Results of ANOVA Kruskal–Wallis H test for the mean inhibitory activity of all the tested honey types and their concentrations against *A. niger*. Statistical differences are indicated by different letters and their significance at *p* ≤ 0.001 with capital letters (A), *p* ≤ 0.05 with lowercase letters (a). Upper and lower case letters above the bars indicate different levels of statistical significance.

**Figure 8 ijerph-20-02458-f008:**
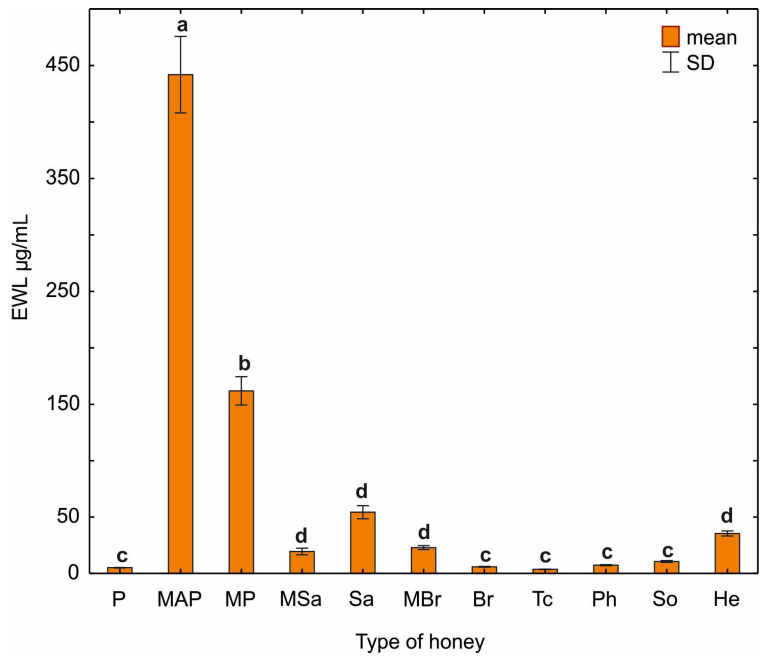
Lysozyme-like activity was determined by the radial diffusion assay and presented as an equivalent of EWL activity (μg/mL). Statistical differences are marked with different letters at *p* ≤ 0.05.

**Figure 9 ijerph-20-02458-f009:**
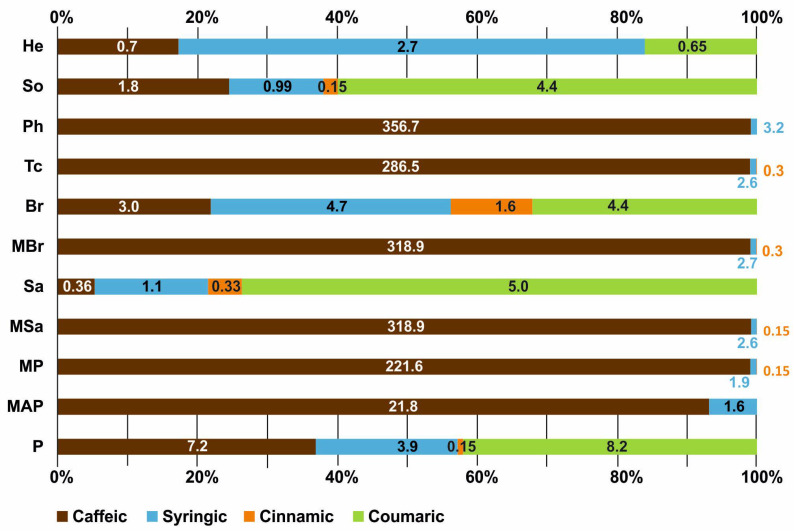
Content of selected phenolic acids (μg/g) in the tested honeys.

**Table 1 ijerph-20-02458-t001:** Physicochemical characteristics of tested honey types, average ± SD, N = 3.

Honey Type	Water Content (%)	pH	Electrical Conductivity (mS/cm)	Total Phenol (μg GAEs/g)	Proteins (mg/mL)
P	16.33 ± 0.47	4.43 ± 0.005	0.29 ± 0.008	670.20 ± 18.96	112.40 ± 2.47
MAP	18.00 ± 0.00	4.62 ± 0.012	0.35 ± 0.009	776.54 ± 9.12	116.80 ± 0.57
MP	14.67 ± 0.47	4.79 ± 0.009	0.29 ± 0.005	700.47 ± 18.18	103.60 ± 0.57
MSa	15.67 ± 0.47	4.68 ± 0.017	0.46 ± 0.014	606.54 ± 27.66	96.80 ± 2.27
Sa	15.00 ± 0.00	4.96 ± 0.029	0.29 ± 0.005	667.14 ± 4.79	107.60 ±0.57
MBr	18.00 ± 0.00	4.67 ± 0.014	0.36 ± 0.012	404.74 ± 9.12	95.20 ± 2.27
Br	17.67 ± 0.47	4.22 ± 0.012	0.27 ± 0.005	378.27 ± 7.30	49.20 ± 3.40
Tc	18.00 ± 0.00	4.08 ± 0.017	0.42 ± 0.012	624.40 ± 15.43	85.60 ± 3.00
Ph	17.67 ± 0.47	4.62 ± 0.005	0.4 ± 0.005	524.40 ± 18.58	90.80 ± 0.57
So	16.33 ± 0.47	4.85 ± 0.009	0.91 ± 0.008	808.05 ± 7.20	85.20 ± 0.00
He	15.67 ± 0.47	4.35 ± 0.012	0.25 ± 0.005	431.27 ± 5.45	41.20 ± 2.47

**Table 2 ijerph-20-02458-t002:** Major sugar components in tested honey samples as determined by HPLC.

Honey Type	Sugars Content (g/100 g), Average ± SD, N = 3						Fructose/ Glucose (Ratio)
	Glucose	Fructose	Sucrose	Rhamnose	Erlose	Fucose	
P	38.80 ± 0.42	39.28 ± 0.12	3.26 ± 0.07	1.18 ± 0.04	3.02 ± 0.10	0.00 ± 0.00	1.01
MAP	44.77 ± 0.10	43.57 ± 0.28	2.42 ± 0.08	1.79 ± 0.07	0.66 ± 0.02	0.23 ± 0.01	0.97
MP	38.80 ± 0.17	38.71 ± 0.23	2.79 ± 0.08	1.24 ± 0.07	2.00 ± 0.07	0.00 ± 0.00	1.00
MSa	40.75 ± 0.21	42.74 ± 0.39	3.08 ± 0.09	1.58 ± 0.04	1.74 ± 0.10	0.00 ± 0.00	1.05
Sa	37.51 ± 0.18	39.24 ± 0.15	3.46 ± 0.07	1.71 ± 0.03	2.92 ± 0.10	0.00 ± 0.00	1.05
MBr	39.56 ± 0.17	42.1 ± 0.58	2.45 ± 0.07	1.01 ± 0.08	1.57 ± 0.06	0.00 ± 0.00	1.06
Br	39.05 ± 0.35	38.53 ± 0.47	4.96 ± 0.08	1.46 ± 0.05	8.26 ± 0.31	0.00 ± 0.00	0.99
Tc	45.10 ± 0.27	38.23 ± 0.28	1.31 ± 0.07	0.00 ± 0.00	0.00 ± 0.00	0.00 ± 0.00	0.85
Ph	52.53 ± 0.46	37.17 ± 0.34	0.00 ± 0.00	0.00 ± 0.00	0.00 ± 0.00	0.00 ± 0.00	0.71
So	39.51 ± 0.35	40.36 ± 0.47	2.89 ± 0.02	1.20 ± 0.05	1.77 ± 0.13	0.00 ± 0.00	1.02
He	37.27 ± 0.29	42.61 ± 0.20	2.91 ± 0.08	1.55 ± 0.02	0.69 ± 0.02	0.35 ± 0.02	1.14

**Table 3 ijerph-20-02458-t003:** Lysozyme-like activity in honey samples.

Honey Sample	Lysozyme-like Activity Expressed as µg/mL EWL
(1) after centrifugation	2.3 ± 0.47
(2) after dialysis	0.71 ± 0.11
(3) after lyophilization	1.5 ± 0.26

**Table 4 ijerph-20-02458-t004:** Selected phenolic acid content; (-)–not detected.

Honey Type	Phenolic Acids (μg/g)			
	Coumaric	Caffeic	Cinnamic	Syringic
P	8.2	7.2	0.15	3.96
MAP	–	210.78	–	1.6
MP	–	221.6	0.15	1.99
MSa	–	318.9	0.15	2.6
Sa	5.0	0.36	0.33	1.1
MBr	–	318.9	0.3	2.7
Br	4.4	3.0	1.6	4.7
Tc	–	286.45	0.3	2.6
Ph	–	356.72	–	3.2
So	4.4	10.8	0.15	0.99
He	0.65	0.7	–	2.7

## Data Availability

All data generated or analyzed during this study are included in this published article. Pollen collections and honey samples are available at the Department of Immunobiology, Maria Curie-Skłodowska University in Lublin, Poland.

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
