# Peer review of "Chemical Composition and Antimicrobial Activity of New Honey Varietals"

_ijerph, 2023, doi:10.3390/ijerph20032458_

Round 1

Reviewer 1 Report

This research reported the honeys and their antimicrobial activities. In this research, the author used 11 honey samples, also the honey was tested against various bacteria. The antibacterial effect is largely related to the amount of phenolic acid and caffeic acid in the honey. In addition, the antifungal activity was also tested to show the applications of these honeys. Overall, this is very interesting research to test common honey with antibacterial effects. More importantly, nowadays, the multidrug resistant and fungal strains is widespread occurrence. This research offers an alternative way, for instance, using natural compounds to obtain the antibacterial effects without causing drug resistance. Some suggestions are listed as below:

1)     Table 1 is so large, removing some information to the supporting information is best.

2)     In the introduction, the author showed this research was aimed to examine of the new varietal honeys with the antibacterial. Thus, the story should start with the urgent need for antimicrobial substances.

3)     Whether there are differences between different brands of honey, the authors need to clarify.

4)     The authors needed to compare other different antimicrobial materials, such as polymers, peptides, and antimicrobial agents to highlight the advantages of honey, especially for drug resistance. Materials Today Chemistry, 2022, 26: 101252. Nano Research, 2022, 15, :5556–5568 

Author Response

Thank you very much for your insightful remarks. Thank you for your time and effort. We made our best to adhere to your suggestions. The latest version of manuscript was proofread by English proofreader.

Some suggestions are listed as below:

  • Table 1 is so large, removing some information to the supporting information is best.

 The whole manuscript was rearranged and many unnecessary details were removed. Additionally, Tables 1 and 3 were removed to the Supplementary materials. The same with Figures 3, 8, 10, 11, 12 – which were removed to the Supplementary materials.

  • In the introduction, the author showed this research was aimed to examine of the new varietal honeys with the antibacterial. Thus, the story should start with the urgent need for antimicrobial substances.

I rearranged the introduction and deleted the first paragraph.

3)     Whether there are differences between different brands of honey, the authors need to clarify.

I have added the necessary data about differences between different brands of honey.

Lines 74-77: These factors differ significantly for different varieties of honey and determine the dif-ferentvarious properties of honeys e.g. Tualang honey ishas been proved to have impact on oral diseases, Acacia honey has antiproliferation potential and manuka honey has the antibacterial activity against Helicobacter pylori [1].

4)     The authors needed to compare other different antimicrobial materials, such as polymers, peptides, and antimicrobial agents to highlight the advantages of honey, especially for drug resistance. Materials Today Chemistry, 2022, 26: 101252. Nano Research, 2022, 15, :5556–5568 

Thank you for pointing out some very interesting publications. I include the information contained in them in the discussion.

Lines 530-531: Recently, there is an urgent need for new substances with antimicrobial capabilities against which pathogenic bacteria and fungi do not develop resistance [1, 2, 52, 53].

Reviewer 2 Report

Dear authors,

The manuscript is interesting. However, I strongly suggest resubmission of the article after careful overhaul of the manuscript. I am making this recommendation on the following reasons: 

(1) Abstract should be clearer and more concise. 

(2) The study objectives should be clear and should be stated specifically, not in broad strokes.

(3) Data could have been analyzed and presented better.

(4) The whole manuscript is too long and so many unnecessary details. I suggest to choose major figures and tables only, or combine some figures that can make one discussion point. 

(5) I found the manner in which the paper was written inconsistently, making it confusing and incoherent.

(6) Minor comment: Some scientific names are not italicized.

Overall, the manuscript has various grammar, spelling, and syntax errors (e.g. in the title: they-> their antimicrobial activities). I had a hard time understanding the manuscript. I recommend having it checked by a professional proof-reader. 

Author Response

Thank you very much for your insightful remarks. Thank you for your time and effort. We made our best to adhere to your suggestions. The latest version of manuscript was proofread by English proofreader.

I am making this recommendation on the following reasons: 

  • Abstract should be clearer and more concise. 

The abstract was corrected and shortened by almost 1/3 - out of 2411 characters with spaces to 1877 characters with spaces.

  • The study objectives should be clear and should be stated specifically, not in broad strokes.

The study objectives were corrected.

Lines 103-108: Due to the broad spectrum of honey properties, this research was aimed  at determining a type of honey, the content of its bioactive substances and its antimicrobial properties against bacteria (such as: Escherichia coli, Bacillus circulans, Staphylococcus aureus, Pseudomonas aeruginosa), yeasts (such as: Saccharomyces cerevisiae and Candida albicans) and mold fungi (such as Aspergillus niger).

  • Data could have been analyzed and presented better.
  • (4) The whole manuscript is too long and so many unnecessary details. I suggest to choose major figures and tables only, or combine some figures that can make one discussion point. 

The whole manuscript was rearranged and many unnecessary details were removed. Additionally, Tables 1 and 3 were removed to the Supplementary materials. The same with Figures 3, 8, 10, 11, 12 – which were removed to the Supplementary materials.

  • I found the manner in which the paper was written inconsistently, making it confusing and incoherent.

The whole manuscript was rearranged and many unnecessary details were removed.

  • Minor comment: Some scientific names are not italicized.

All scientific names were checked and corrected if it was necessary.

Overall, the manuscript has various grammar, spelling, and syntax errors (e.g. in the title: they-> their antimicrobial activities). I had a hard time understanding the manuscript. I recommend having it checked by a professional proof-reader. 

The latest version of manuscript was proofread by English proofreader.

Reviewer 3 Report

The work is very interesting but the english grammar and style is distracting. Revamping the figures could provide more clarity to the reader.

Title Suggestion: Chemical Composition and Antimicrobial Activity of New Honey Varietals

English grammar and spelling should be revisited throughout the manuscript since it interferes with understanding and engagement of the reader. Some small fixes:

·       ‘the’ and other articles are missing or superfluous

·       spelling of some of the keywords

·       spelling of physicochemical

·       limit use of ‘it’ and other pronouns

·       correct ‘they’ to ‘their’

·       paragraph structure throughout

·       transition words and phrases should be pared down or eliminated unless required for clarity

Rewrite the Results section to align format, improve readability, and explain terms. Reorganize the bulleted list under the Mellisopalynogical Analysis section (2.1) so that it is more paragraph-like or made into a table. Remove numbered and bulleted lists and replace them with paragraph-style lists throughout.

Split Table 1 into two tables, Table 1 for varietals and Table 2 for multiflorals, could make smaller and more informational tables. Current Table 1 is cumbersome and overwhelming to the reader.

Consider adding a bar graph for the phenolics content data separately to highlight it.

While ordination diagrams can be useful representations of data, they do not appear to demonstrate this data effectively and the description of these figures in the text must be rewritten for clarity.

The current Tables 2, 3, and 4 should be reworked so as NOT to include:

·       the 0.00 + 0.00 notation when “not detected” is more accurate

·       the European decimal notation

·       more decimal places in the presented numbers than can be justified

The remaining sections of the Results include figures that lack either clarity, units, or both. These figures should be redrafted for clarity. The catalase data does not appear to be necessary or particularly enlightening to justify three figures.

The tables in the Materials and Methods section should perhaps be moved to Supplementary Materials as reference tables.

Author Response

Thank you very much for your insightful remarks. Thank you for your time and effort. We made our best to adhere to your suggestions. The latest version of manuscript was proofread by English proofreader.

Title Suggestion: Chemical Composition and Antimicrobial Activity of New Honey Varietals

Thank you very much. The title was changed as you pointed.

English grammar and spelling should be revisited throughout the manuscript since it interferes with understanding and engagement of the reader.

The latest version of manuscript was proofread by English proofreader.

 Some small fixes:

  • ‘the’ and other articles are missing or superfluous
  • spelling of some of the keywords
  • spelling of physicochemical
  • limit use of ‘it’ and other pronouns
  • correct ‘they’ to ‘their’
  • paragraph structure throughout
  • transition words and phrases should be pared down or eliminated unless required for clarity

 Thank you very much for pointing all the mistakes. The latest version of manuscript was proofread by English proofreader.

Rewrite the Results section to align format, improve readability, and explain terms. Reorganize the bulleted list under the Mellisopalynogical Analysis section (2.1) so that it is more paragraph-like or made into a table. Remove numbered and bulleted lists and replace them with paragraph-style lists throu

The Results section was corrected and changed in  more paragraph-like style.

Lines 111-130

On the basis of mellisopalynogical analysis, 7 varietal honeys and 4 multifloral honeys were distinguished. Among the varietal honeys, the following were distinguished: plum honey (P) with the Prunus type (46.98%)dominant pollen; willow honey (Sa) with the Salix sp. (70.25%) dominant pollen; rapeseed honey (Br) with the Brassicaceae type (81.70%) dominant pollen ; lime honey (Tc) with the Tilia sp. (28.99%) dominant pollen; Phacelia honey (Ph) with the Phacelia thanacetifolia (65.62%) dominant pollen; honeydew honey (So) with the Solidago type (46.48%) dominant pollen; and sunflower honey (He) with the Helianthus type (73.35%) dominant pollen (Supplementary materials Table S1 and S2).

In the case of multifloral honeys, they were characterized as follows: multifloral-Br (MBr) with the predominance of pollen from Brassicaceae type (33.01%), Aesculus hippocastanum (15.53%) and Poligonum bistorta (15.53%); multifloral-Sa (MSa) with the predominance of pollen from Salix sp. (21.55%) Solidago type (17.24%) and Tilia sp. (17.24%); multifloral-AP (MAP) with the predominance of pollen from Acer sp. (37.38%) and Prunus type (37.38%); and multifloral-P (MP) with the predominance of pollen from Prunus type (29.47%), Brassicaceae type (15.79%) and Salix sp.(15.26%)(Supplementary materials Table S1 and S2).

Split Table 1 into two tables, Table 1 for varietals and Table 2 for multiflorals, could make smaller and more informational tables. Current Table 1 is cumbersome and overwhelming to the reader.

 The whole manuscript was rearranged and many unnecessary details were removed. Additionally, Tables 1 and 3 were removed to the Supplementary materials. The same with Figures 3, 8, 10, 11, 12 – which were removed to the Supplementary materials.

Consider adding a bar graph for the phenolics content data separately to highlight it.

 A graph was added as Figure 9.

While ordination diagrams can be useful representations of data, they do not appear to demonstrate this data effectively and the description of these figures in the text must be rewritten for clarity.

All descriptions were corrected.

The current Tables 2, 3, and 4 should be reworked so as NOT to include:

  • the 0.00 + 0.00 notation when “not detected” is more accurate
  • the European decimal notation
  • more decimal places in the presented numbers than can be justified

 All Tables were corrected.

The remaining sections of the Results include figures that lack either clarity, units, or both. These figures should be redrafted for clarity. The catalase data does not appear to be necessary or particularly enlightening to justify three figures.

 All Figures were corrected. Additionally, Figures 3, 8, 10, 11, 12 – were removed to the Supplementary materials.

The tables in the Materials and Methods section should perhaps be moved to Supplementary Materials as reference tables.

 The whole manuscript was rearranged and many unnecessary details were removed. Additionally, Tables 1 and 3 were removed to the Supplementary materials. The same with Figures 3, 8, 10, 11, 12 – which were removed to the Supplementary materials.

Round 2

Reviewer 2 Report

Kindly check some minor typos  (eg. S. aures in Abstract) and un-italized Scientific Names (eg. Fig 5 & 6) all over the manuscript.

If you can narrow down the number of Tables, Figures, and Supplementary figures. You do not need to show all. 

Author Response

Kindly check some minor typos  (eg. S. aures in Abstract) and un-italized Scientific Names (eg. Fig 5 & 6) all over the manuscript.

If you can narrow down the number of Tables, Figures, and Supplementary figures. You do not need to show all. 

Thank you very much for your insightful remarks. Thank you for your time and effort. We made our best to adhere to your suggestions.

Manuscript was checked thoroughly throughout (especially the Abstract). Scientific Names on Figures were un-italicized all over the manuscript.